# BALANCE BEAM: ADAPTIVE COMPUTATION FOR AFFORDABLE TRAINING AND INFERENCE WITH HIGH-THROUGHPUT OFFLOADING FOR LLMS

## ABSTRACT

With the surging growth of model parameters, foundation models pose unprecedented challenges to traditional computational infrastructure. These large models intrinsically require substantial accelerator memory to accommodate massive tensors including model weights, activations, and optimizer states during pretraining, fine-tuning or even inference stages. To alleviate such intense pressure on memory, besides introducing excessive accelerators to suffice high demand of memory, offloading these parameters from accelerator to other storage medium such as DRAM is a preferable option for fine-tuning or inference with the model under computationally restricted circumstances. However, the prohibitive costs of data movement render it a theoretically plausible yet practically unpreferred solution. Previously state-of-the-art methodologies enhanced inference performance by retaining partial model state in-situ across multiple mini batches to boost inference performance but incur intricate hyperparameters and excessive overhead of exchanging cache. In this work, we propose a comprehensive workflow to address these challenges, with focuses on dynamic analysis of model-system compatibility and prioritizing computational intensity over data movement. We have shown that the proposed workflow facilitates both fine-tuning and inference of foundation models with higher throughput in restricted computational resources. Compared to state-of-the-art approach, our framework attains a remarkable speedup of over 4x for training and 2x for inference, using a 30-billion parameter model on a singular NVIDIA A100 GPU.

## 1 INTRODUCTION

With the emergence of large language models (LLMs) such as GPT (Brown et al., 2020; OpenAI, 2023), LLaMa (Touvron et al., 2023a;b), PaLM (Chowdhery et al., 2022), etc., the capabilities of LLMs have been continuously growing alongside the increase in model sizes, which exponentially increases in recent years (De Angelis et al., 2023). The continuous evolution of LLMs has brought about severe challenges in model training due to massive parameters such as model weights, activations, gradients, and optimizer states. For example, training GPU-4 requires a computing cluster of 25,000 NVIDIA A100 GPUs, and it takes around 90-100 days to reach an acceptable convergence of gradient descent, costing tens of millions of dollars, which have far exceeded what small enterprises and AI application developers can afford. In order to alleviate the enormous cost of training of LLMs, offloading technologies that refer to the practice of moving or distributing parts of training data or processes to other resources or hardware components was proposed. Zero Redundancy Optimizer(ZeRO)-Offload introduced by DeepSpeed allows offloading optimizer states from GPU memory to CPU RAM for mitigating the memory and computational demands of training large models while using fewer GPUs (Ren et al., 2021). ZeRO-Infinity, as an extension of ZeRO-Offload, was proposed to offload more data to CPU and/or NVMe memories with overlapping computation and communications for even higher bandwidth utilization (Rajbhandari et al., 2021). In the context of inference, offloading can also be utilized to significantly reduce the computing resource required for pre-trained models at unprecedented scale (Aminabadi et al., 2022). Recently, researchers (Sheng et al., 2023) have shown that high-throughput generative inference of a LLM with 175 billion parameters is feasible with a single GPU thanks to an efficient offloading strategy.

While offloading technology can substantially conserve memory and computational resources when enhancing the scalability of foundation models during both training and inference phases, it inevitably introduces some challenges. One of these challenges pertains to the communication overhead incurred when transferring model parameters between memory mediums, which may result in an extension of the time required for model training and inference. For instance, model weights need to be frequently shuttled between GPU memory and CPU memory during pre-training or fine-turning. In phase of inference, a large volume of activations including Key Value cache (KV Cache) (Pope et al., 2022) needs to be moved to external storage to alleviate the strain on GPU memory, and subsequently brought back to GPU for next round of computation. In fact, the latency consumed by data movement surpasses the computation time of models due to the bandwidth bottleneck between GPU, CPU and NVMe storage, which brings in a high latency to the task. For instance, the FlexGen proposed in (Sheng et al., 2023) have showed that a total latency of 5000 s is required to generate 2,048 tokens for OPT-175B model with a computing node (1x NVIDIA T4 16GB GPU, 208GB CPU DRAM and 1.5TB NVMe SSD), where the average latency is around 2.44 s/token. Such level of performance is far from the computation capability of the given system. Hence, it is important to optimize the trade-off between computation and data movement when applying offloading given restricted computational resources.

Another challenge of offloading lies in configuration complexity that refers to the strategy of configuring the hyperparameters by which both computing and communication could be fully overlapped for optimal performance of offloading (Wang et al., 2023). It should be noted that since the configuration of hyperparameter intricately depends on the model architectures and computing hardware, improper selection of hyperparameter can lead to system instability and severe performance degradation of LLMs. For instance, a large batch size can result in excessively large KV cache during computation, causing inference tasks to be terminated prematurely due to insufficient memory. Therefore, the selection of hyperparameters in offloading appropriately is crucial for improving the efficiency of data movement, thereby optimizing the performance of LLM tasks. However, there is no one-size-fits-all approach for hyperparameter configuration, and most of the previous work about hyperparameter configuration in offloading was based on empirical knowledge or preliminary search. It is more than a necessity to have an adaptive strategy of configuring hyperparameters in the practice of offloading for LLMs.

In order to address the aforementioned challenges, in this paper, we propose *Balance Beam*, an workflow to optimize the trade-off between latency and throughput performance of LLMs when offloading applied. We first give a comprehensive analysis on the issues of current approaches of offloading from perspectives of computation and data movement, and then introduce how Balance Beam addresses the issues through a serial of strategies including hyperparameter selection, KV cache management, and asynchronous tensor exchange. We evaluate the performance of Balance Beam with LLMs including OPT-6.7B, OPT-30B, and OPT-175B, and results have shown that Balance Beam can not only be applied for inference task with 2x speedup, but also training task with over 4x speedup with a single NVIDIA A100 GPU, compared to state-of-the-art (SOTA) approaches. The main contribution of this work includes as follows:

**First**, we discover that the time taken by moving model parameters is significantly longer than that for training data, and this component consumes a substantial portion of each iteration. Thus we carefully design an optimized data movement strategy combined with improved forward and backward computation methods to reduce the frequency of reading model weights from CPU memory into GPU memory, leading to a significant improvement on training efficiency.

**Second**, taking KV caching, batch size, number of gradient checkpoints into account, we propose a balancing strategy to simplify hyperparamenter selection for both training and inference tasks. We then extensively demonstrate the efficiency of the proposed hyperparameter selection strategy and show that it achieves best performance compared with existing schemes.

**Last but not least**, unlike FlexGen (Sheng et al., 2023) where latency performance is traded for high-throughput inference scenarios, our proposal focuses on maximizing the utilization of hardware resources by analyzing the model-system compatibility to optimize data movements and computation tasks for minimal overhead in offloading. It provides an effective framework to conduct LLMs tasks (training/fine-tuning and inference) even with extremely limited computation resources.

## 2 BALANCE BEAM'S OVERVIEW

This section gives an overview of the proposed Balance Beam workflow. We have provided an in-depth description of the computational graph employed in the scheme during both training and inference phases, along with a data movement analysis to underscore the core concept.

### 2.1 COMPUTATIONAL GRAPH OF BALANCE BEAM

Computation with transformer-family LLMs (Vaswani et al., 2017) can be formulated as a graph traversal problem, where model and data are partitioned into segments that can fit within the accelerator's memory capacity without overflowing. In canonical solutions, each inference iteration processes a single batch (i.e., traverse the graph row by row), and a complete traversal of the model is required for each new token. This approach demands high I/O amounts between host and accelerator to manage the model weights, and it necessitates reading the entire model for each new token. Alternatively, we construct a column-wise traversal path similar to the solution presented in Sheng et al. (2023) (see Fig. 1(a)). In the figure, solid lines denote computation path of forward propagation, while dashed lines signify backward propagation.

**Effective inference.** A large batch, namely an effective batch, is split into sub batches, with the parameters for each model block being preserved *in-situ* over multiple sub batches. In each sub-batch, the associated activations and KV cache are immediately offloaded to prevent memory spikes on the accelerator. These states are then reloaded onto the accelerator prior to the computation of next adjacent block. Therefore, peak memory consumption is effectively alleviated by retaining only several layers of KV cache on the accelerator, and the I/O demands can be significantly reduced by loading the model weights only once across multiple sub-batches.

**Checkpoint-based training.** The traversal strategy has also been extended to the training phase, with a distinctive I/O pattern that involves bulky transfer of activations and gradients. Firstly, KV caching is not activated during training and activations are retained for subsequent gradient computations. Secondly, gradients are traced back to update corresponding model parameters during the backward propagation of training phase. These factors collectively exert substantial memory pressure on the accelerator. To address the challenges, we have implemented gradient checkpointing. Only one copy of the input tensors is saved for each block on the graph, and other intermediate activations are recomputed as needed during backward propagation. Furthermore, each hidden state between the blocks is immediately offloaded upon completion of computation. The offloading implementations and schedule will be discussed later in Section 3.2.

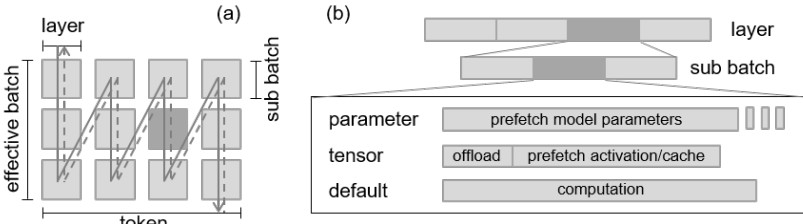

Figure 1: **(a)** Computational graph of Balance Beam. **(b)** Asynchronous execution streams.

### 2.2 DATA MOVEMENT ANALYSIS

To achieve optimal training and inference throughput, we aim to strike a balance between minimizing data movement and maximizing arithmetic density iterating through the same amount of training data. In this context, we will discuss scenarios where KV caching is enabled for inference and gradient checkpointing is enabled for training.

Data movements in a vanilla training loop are illustrated in Fig. 2. In forward propagation stage, for each layer, model weights are loaded into device and the activations are written to host once (In inference, KV cache between layers is written). In backward propagation stage, the model weights together with activations are loaded from host to devices. After backward propagation of each

module, corresponding gradients are written back to host for parameter update. Note that only one mini-batch is processed in the pipeline, and batch size is memory-bounded.

Balance Beam, in contrast, handles multiple sub batches in one iteration. Model weights are transferred just once for a single effective batch, and subsequently reused across multiple sub-batches. To alleviate memory pressure, we offload and reload activations or KV cache for each sub-batch in training and inference, respectively. In training, the host-to-device data transfer for our approach $D_{hd}$ and vanilla approach $D'_{hd}$ are listed as below, where $h$ refers to model hidden size, $n$ is the maximal token number that accelerator can hold in one iteration and $N$ denotes the total token number in the effective batch:

$$D_{hd} = 24h^2 + 2Nh, D'_{hd} = \frac{N}{n}(24h^2 + nh)$$

With the data movement clear in the loop, the elimination point of activation read overhead, $D_{hd} < D'_{hd}$, can be easily computed (details see App. A.1):

$$N > \frac{24nh^2}{24h^2 - nh}$$

Above gives us a theoretical lower bound of effective batch size that guarantees elimination of overhead. Importantly, this suggests that with larger effective batch size, the overhead of fetching model parameters can be asymptotically hindered. However, the equilibrium comes earlier in actual computation due to overlapped communication and distinct hardware characteristics. Therefore, we do not tune hyperparameters merely on the data movement, but also take real-world runtime performance into account.

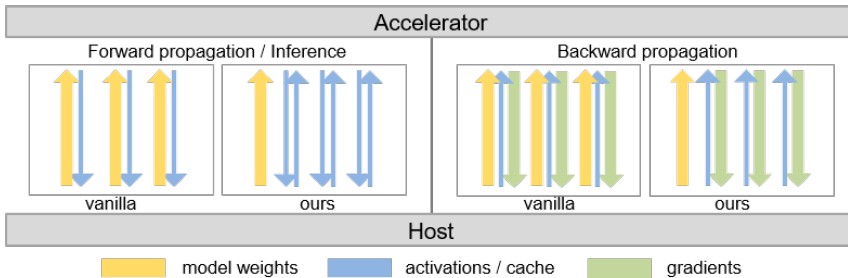

Figure 2: Data movements with offloading in LLM training/inference of an effective batch.

## 3 METHODS

We describe hyperparameter balancing strategies and optimized offloading scheme in this section. These technologies are applicable to both the training and inference phases, with specific adjustments detailed for each phase as outlined below.

### 3.1 HYPERPARAMETER BALANCING STRATEGIES

Hyperparameter selection is a critical factor in both the training and inference phases of LLMs. However, it often goes unaddressed in the literature, as it is typically assumed that researchers will default to choosing the most suitable hyperparameters based on preliminary searches and empirical observations. In this study, we take the pioneering step of theoretically analyzing the hyperparameter selection process, which we refer to as 'balancing strategies'. We also provide experimental validation in Section 5.1, demonstrating how these hyperparameters can be carefully balanced to achieve the highest throughput while considering various trade-offs.

**KV caching.** While KV caching can significantly boost inference computation, it also introduces substantial memory and I/O pressure. It is well-established that a larger batch size can enhance throughput because the overhead of prefetching time is offset by computation. However, the pinned host memory for caches imposes an inherent upper limit on batch size. Therefore, the optimal throughput can only be achieved by striking a balance between the advantages of a large batch size

and the corresponding KV cache. If memory fails to accommodate a single sub-batch or if the KV cache significantly impacts batch size selection, particularly with extremely large models, it is advisable to disable KV caching. To simplify this balancing strategy, our utility will recommend a minimum batch size for achieving high throughput, and caching will not be employed if this threshold cannot be met.

**Batch size.** Aligned sub-batch size is a key hyperparameter impacting computation. Major arithmetic pressure of transformers lies in Linear layers, where batched general matrix-matrix multiplication (GEMM) is heavily used. Depending on different implementation of accelerator backends, performance can largely benefit from proper batch size (Abdelfattah et al., 2016). Take NVIDIA GPU as an example, according to vendor guidelines, achieving optimal half precision computation entails ensuring that input sub-batch sizes within model layers are multiples of 8. In the particular case of A100, it is advisable to keep batch size a multiple of 64. Moreover, for larger batches exceeding 96, it's advisable to use sizes that are powers of 2. In addition to aligning sub-batch sizes, enlarging sub-batch sizes and effective batch sizes are preferred due to the substantial benefits they offer during both the training and inference phases. To fully harness the potential of the hardware and strike the right balance with batch size configuration, our workflow selects appropriate batch sizes by assessing the working memory requirements per token during benchmarking.

**Number of gradient checkpoints.** The checkpointing mechanism is a widely employed technique in LLM training, serving to conserve memory or enable the resumption of training from saved states. Having more layers between two checkpoints results in fewer hidden states to be saved, thereby creating room for a larger effective batch size. Throughput can benefit from the increment of both parameters (i.e., number of checkpoints and effective batch size), but they cannot be scaled up simultaneously. Therefore, leveraging the runtime characteristics gathered during the analysis phase, we assess the gain of increasing checkpoints and effective batch size and this evaluation enables the utility to determine an optimal balance between these two parameters. It is important to highlight that the number of live layers also serves as an accelerator memory-bound hyperparameter during the inference phase but shows marginal impact on throughput. Consequently, we have implemented a strategy to retain only one layer on the accelerator at any given time during inference.

## 3.2 OPTIMIZED OFFLOADING SCHEME

Offloading technology has significantly mitigated hardware memory stress during both LLM training and inference, albeit with the inherent introduction of additional I/O costs. In the present study, we have devised an optimized offloading scheme to prevent memory peaks on the accelerator and to compensate for the supplementary communication overhead. In this scheme, activations and KV cache are strategically offloaded to facilitate the execution of larger models. The efficient management of bulk host-accelerator tensor exchanges is achieved through our comprehensive asynchronous tensor swapper, enabling seamless overlap between prefetching, offloading, and computation processes.

**Contiguous buffer pool.** PyTorch adopts an intuitive approach to allocate buffers for offloaded tensors, where a new buffer is allocated every time a new tensor arrives. However, allocating extensive continuous memory in this manner is a lengthy and blocking operation. Frequent allocations, as in the case of PyTorch, can result in unnecessary time wastage and hinder complete asynchronous overlapping. In our scheme, the swapper takes a different approach by establishing a fixed buffer pool specifically designated to receive and store tensors offloaded from accelerators. This proactive step, initializing a contiguous buffer pool before the commencement of computation loops, can significantly mitigate computational delays.

**Asynchronous tensor exchange.** Our scheme employs an asynchronous tensor exchange strategy to overlap the prefetch of the next sub-batch, the computation of the current sub-batch, and the offload of the last sub-batch (see Fig. 1(b)). Prior to initiating the forward computation of each sub-batch, the swapper submits prefetching and offloading operations to separate CUDA streams. Upon completion of the computation for each sub-batch, the streams are synchronized to ensure the availability of the next hidden state and KV cache.

**Integrated activation manager.** Throughout the training process, the tensor swapper not only preserves the activation of the current layer but also archives previous activations at each gradient checkpoint. A context manager monitors saved tensor for backward propagation, which provides

deep learning framework with precise references to the saved tensor and concurrently the prefetching and offloading operations. This systematic approach ensures seamless integration and fluid interaction between the stored tensors and the computational components, facilitating optimal execution of computation and overall throughput.

**Fine-grained KV cache offloading.** KV cache within the same sub-batch is preserved for each layer throughout the generative steps to maintain computational consistency, which can be considered as a layer-wise approach. However, it is also possible to optimize KV cache offloading with a fine-grained approach, utilizing token-wise application to minimize redundancy.In this method, only the KV state of the new token is written to the host memory. To ensure the continuity of host pinned memory buffer, the buffer pool is organized in the format of (num sub batches,layer,sequence length,-1). Given the augmenting nature of sequence length through generation steps, it is permuted to the third dimension from its original position in the sixth dimension (Fig. 3). Therefore, a contiguous memory chunk is available for asynchronous copy for each token in the sub batch, eliminating the need for blocking or additional loops. This meticulous scheduling minimizes costly writing operations, preserving sufficient resources for efficient computations and ultimately enhancing the overall arithmetic intensity.

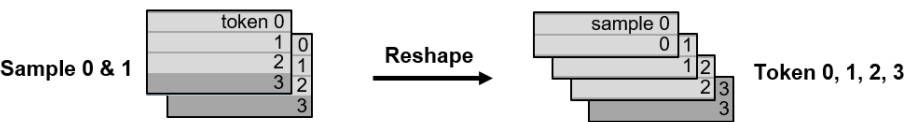

Figure 3: Reorganizing buffer memory to ensure continuity.

# 4 EVALUATION

**Implementation.** For any transformer-family model, we provide an abstract mix-in class that enables the model to compute and offload activations of several sub-batches without peaking in device memory. The implementation also comes with utilities that automatically tune the hyperparameters. Our current implementations include support of both OPT (Zhang et al., 2022) and LLaMA (Touvron et al., 2023a;b), while other transformer-based models can be applied with minimal efforts. For stringent comparison, we picked the OPT-6.7B, OPT-30B, and OPT-175B for benchmarking. The model attention layer is replaced by layers with fused QKV dot product attention (Dao, 2023; Dao et al., 2022) from optimum package.

For optimal offloading and scalability, we employed DeepSpeed with ZeRO-Offload stage-3 (Rajbhandari et al., 2019; 2021; Ren et al., 2021). A dedicated model engine manages tensor placement, communication and distribution, ensuring organized and coherent model operation. Half precision training/inferencing with 16-bit brain floating numbers is enabled to minimize memory pressure and maximally utilize tensor cores on NVIDIA GPU.

**Benchmark pipeline.** We designed meticulous benchmark pipelines to characterize different runtime performance of our approach and establish baselines. To quantify impacts of hyperparameters, we examine the different combinations effective batch size, sub batch size, number of layers between checkpoints, and sequence lengths. After a coarse-grained experiment, we confirmed the independence between hyperparameter impacts, and therefore proceeded with single variable experiments. For optimal performance benchmarking, we provide fixed-length inputs to the models for identical iterations, comparing our approach with SOTA and baseline solutions. For inference, we record the elapsed time of prefilling (i.e., build trace and generate KV cache) and decoding steps. Hence, we can get the throughput out of batch size and elapsed times; For training, we examine the floating-point operations per second (FLOPS) to quantify the arithmetic intensity of training.

**Platform specifications.** Our experiments are performed on nodes from a high-performance cluster. Our benchmark jobs usually request 4 NVIDIA A100-40GB PCIe GPUs with NVLink. For each GPU, the scheduler also assigns 16 cores from AMD EPYC 7713 and 160GB RAM on the nodes. Each node is equipped with a local 14 TB NVMe RAID. For scalability task, the nodes are connected via 100 Gbps EDR Infiniband-over-Ethernet connections.

# 5 RESULTS

## 5.1 IMPACTS OF HYPERPARAMETERS

As an experimental basis of hyperparameter tuning, we first analyzed the impact of changing hyperparameters (Fig. 4). For the sake of computing time, we tested the hyperparameter settings under model size of 6.7 billion parameters.

Batch size influences both training and inference in a way that largely aligns with our expectations: A hyperbolically ascending curve. However, we have observed that the relationship between the effective batch size and training FLOPS is not strictly monotonic. There exists an intrinsic threshold in terms of token numbers, beyond which a larger effective batch size can adversely impact arithmetic intensity. This threshold is determined during the analysis phase, thereby influencing batch size decisions based on both the threshold and the available host memory. We suspect this behaviour is highly associated with the gradient partition and reduction pattern that managed by ZeRO mechanisms. However, further investigation is out of the scope of this study. We leverage the observed training token threshold to set the upper limit of effective batch size in our analyse phase.

Enlarging of sub batch size also hyperbolically facilitate the increment of arithmetic intensities. But in the case of inference, the throughput reaches a stable plateau after 64 samples per sub-batch. This is within our expectation: Although increment of sub-batch size substantially compresses computation time per token, the cost of loading KV cache is also rapidly expanding. Due to our full asynchronous execution (Fig. 1b), the shrinkage of computation results in the turnaround awaiting KV cache exchange, thereby limiting the gain of large sub-batch size.

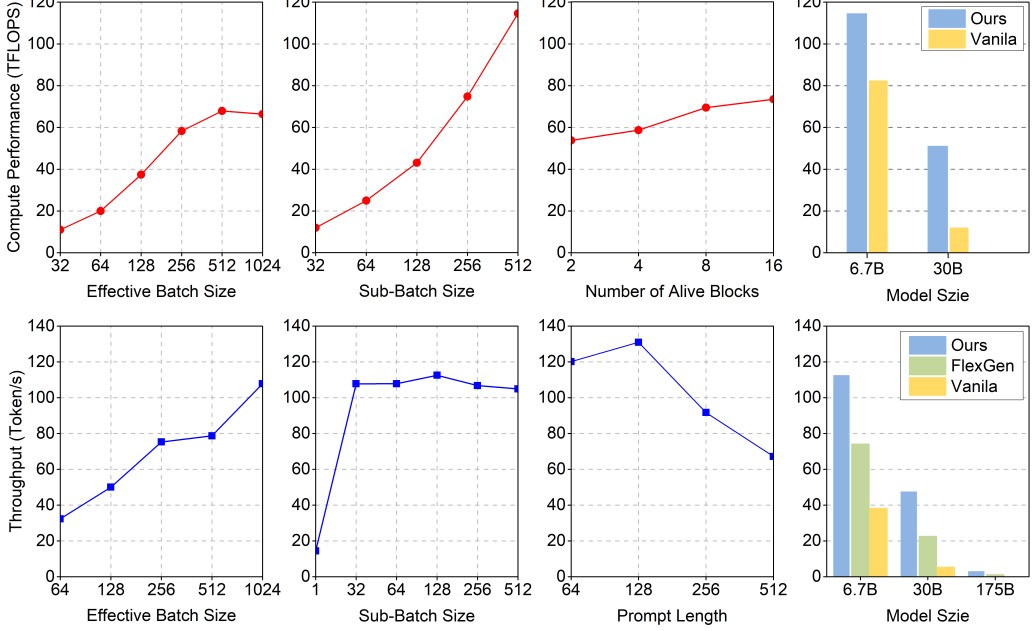

Figure 4: Results from our experiments. First row shows the compute performance during fine-tuning; Second row shows the throughput during inference. The line plots shows the impact of changing hyperparameters to model performance; The bar plot shows the comparison between optimal results from our approach and SOTA solutions.

## 5.2 SINGLE DEVICE BENCHMARKING

We comprehensively tested the model performance under optimal hyperparameters, and the results exhibited substantial advantages over the previously SOTA approaches. For fine-tuning, the OPT-175B is excluded due to extensive memory demands and unrealistic application scenario for single accelerator.

**Fine-tuning.** Our approach substantially levelled-up the fine-tuning throughput in comparison to vanilla offloading scheme (Fig. 4). With optimal hyperparameter settings, our approach achieved an arithmetic intensity of 114.58 TFLOPS on OPT-6.7B, while vanilla approach was only able to maintain a throughput of 82.39 TFLOPS. The performance disparity becomes more pronounced in larger models. For OPT-30B, our approach has an arithmetic intensity of 51.09 TFLOPS, which is in stark contrast to 12.01 TFLOPS of previously acknowledged SOTA performance (speedup around 4.25x, see App. A.2). Substantially elevated arithmetic intensity underscores improved utilization of accelerator devices, a condition under which less time is allocated to turnaround, awaiting I/O operations.

**Inference.** Our approach surpasses the SOTA solution and the baseline at different scale (Fig. 4) during inference phase. In the case of OPT-6.7B, our solution has a maximum 112.52 tokens per second (Token/s), while FlexGen and ZeRO stage-3 maintain 74.26 and 38.39 Token/s under similar hyperparameters, respectively. This advantage is also extrapolated to superior performances in larger models: For OPT-30B and OPT-175, our solution attains speedup ratios of 2.09 and 2.35 compared to FlexGen, and 8.63 and 22.60 in comparison to the baseline, respectively.

Note here according to our strategy, KV caching is disabled for OPT-175B, which allowed us to scale our effective batch size to 1024. For OPT-175B, KV cache for each sample with 256 tokens can take up 1.125GB host memory, resulting in heavy I/O overhead and becoming a significant bottleneck to maximize effective batch size. Whereas our strategy without KV caching, can benefit more from larger sub batch size and gains enhanced arithmetic intensity.

## 5.3 SCALABILITY

We also examined the scalability of our approach (Tab. 1). And surprisingly, we identified a super linear scalability in single-node configuration. With per GPU batch size of 512, our solution attained 91.07, 195.19, and 369.74 Token/s on 1, 2, and 4 GPUs, respectively. In terms of per GPU throughput, the 2-GPU and 4-GPU configurations have 97.60 and 92.43 Token/s, respectively. This increment of per GPU throughput is potentially brought by the memory centric design of ZeRO-Offload stage-3, which allows each device to concurrently fetch partitions of model parameter and gather the entire parameter by collective peer-to-peer communication. Additionally, we also tested the multi-node ZeRO parallel scenario. The experiment is conducted on two HPC nodes with Infini-Band EDR interconnectivity (100Gbps). With 1 GPU per node, inter-node scenario achieves a per GPU throughput of 87.51 Token/s. This is a lower gain compared to inner-node scenario, but it is still a solid improvement compared to single-device inference.

## 5.4 ABLATION STUDY

To investigate the effect of each optimization, we conducted a series of ablation studies that opting out certain optimizations (Tab. 2). The flash attention mechanism showed marginal improvements in our settings, yielding approximately a 1.3% increment in throughput. Conversely, we discerned that our smart scheduling of bulk tensor exchange precipitated a substantial throughput augmentation, accounting for a 33.8% increment. Finally, the implementation of engine-managed tensor prefetching further contributed to a 3.9% increment in throughput. These insights are pivotal, shedding light on the nuanced interplay of various optimizations and their cumulative effect on enhancing computational throughput.

| | 1 GPU | 2 GPUs | 4 GPUs | | Decode | Total |
|---|---|---|---|---|---|---|
| **Local** | 91.07 | 195.19 | 369.74 | **Optimal** | **126.92** | **90.33** |
| **Distributed** | N/A | 175.02 | N/A | **w/o Flash Attention** | 127.58 | 89.16 |
| | | | | **w/o Param Prefetch** | 118.97 | 86.95 |
| | | | | **Blocking Caching** | 90.65 | 67.50 |

Table 1: Scalability test (Token/s)  Table 2: Ablation study (Token/s)

## 6 FUTURE WORKS

### 6.1 LARGE BATCH TRAINING

Our approach has demonstrated high throughput in training/fine-tuning scenarios and is fundamentally equivalent to scaling up the mini-batch size, effectively making large-batch training accessible to infrastructure-constrained scenarios. Similar to many scenarios involving large-batch training, fewer model updates demand higher learning rate to maintain training efficiency (You et al., 2017). However, scaling factors for learning rate can vary significantly across different use cases. Although various learning rate scheduling strategies may enhance the adaptability of learning rate, excessively large learning rates can potentially introduce instability in convergence and accuracy (Shallue et al., 2019; You et al., 2017). Recent advances also suggests that normalized gradients can assist in preserving accuracy during large-batch training while concurrently enhancing efficiency. For instance, optimizers like LARS (You et al., 2017) or LAMB (You et al., 2019) provide paradigms of large-batch training with layer-wise regularization. Recent work by Zheng et al. (2020) has proposed block-wise regularization strategies to further improve the efficiency.

To fully harness the potential advantages of our solution, it is essential to incorporate suitable large-batch training strategies. By implementing a meticulously designed learning rate schedule, optimizers, and other advanced techniques, training efficiency can far exceed that of canonical setups. We look forward to the possibility of undertaking this aspect of the work in the near future, with the aim of establishing a robust framework that maximizes both training accuracy and efficiency, even in resource-constrained environments.

### 6.2 DYNAMIC BATCHING

Grouping samples with analogous lengths through bucketing can significantly enhance computational efficiency by reducing the number of padding operations required and eliminating the subsequent computations associated with padding tokens. Tools like the *BucketIterator()* function from the *torchtext* package (Paszke et al., 2019) can minimize amount of padding needed while producing freshly shuffled batches for each new epoch. However, alterations in sequence length directly impact the memory requirements per sample, introducing a potential risk of memory overflow. To make optimal use of available accelerator and host memory while also mitigating the risk of overflow, it's essential to dynamically adjust effective batch sizes whenever a change in sequence length is detected.

The challenge arises due to the fact that altering the number of sub-batches leads to modifications in the established computational graph. This graph is not solely managed by PyTorch but is also recorded by offloading frameworks like DeepSpeed to coordinate model weight prefetching. Post-hoc modification of the computation graph is not officially supported by DeepSpeed, and user-driven modifications may jeopardize the forward compatibility of the current approach. In the future, we anticipate contributing to open-source accelerated deep learning systems with offloading like DeepSpeed (Rasley et al., 2020) and Colossal-AI (Bian et al., 2021), to allow flexible computation strategies.

## 7 CONCLUSION

Our solution manifested SOTA performance in both fine-tuning and inference of LLMs with offloading technology. By striking meticulous balance hyperparameters and implementing various optimizations, this approach empowers users to fully harness the potential of their systems, even within resource-constrained environments.

## 8 REPRODUCIBILITY STATEMENTS

To facilitate reproduction of experimental results, we provide a deployment disk image on Google Cloud. Readers can reproduce part of aforementioned results with image *balance-beam-gcp* and single instance of *a2-highgpu-1g*. The image contains instructions for reproducing the result. For larger models or scalability test, readers may have to seek for platforms with higher capability.

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

## A  APPENDIX

### A.1  DETAILS ON DATA MOVEMENT

This section will give more quantitative details on the training and inference loop data movements.
**Training.** As shown in Fig. 2, device-to-host writes include the activation of last sub batch, gradients; host-to-device reads includes next layer weights and input activation of next sub batch:

$$D_{hd} = 2D_p + 2D_{act} = 24h^2 + 2Nh,$$
$$D_{dh} = D_g + D_{act} = 12h^2 + Nh$$

Where $D_p$, $D_g$, $D_{act}$ denotes data size of parameters, gradients, and activation tensors, respectively.

Typical self-attention layers are composed of 6 essential layers (Vaswani et al., 2017): 4 fully-connected layer for Q, K, V and output projection, and 2 feed-forward layers. Each projection matrix has a weight size of $h^2$, where $h$ is the hidden size of model; Each feed-forward layer generally have 3 times larger size than projection matrix. Additionally, Gradient size is theoretically equivalent to parameter size. Therefore we have:

$$D_p = D_g = 4D_{proj} + 2D_{ffn} = 12h^2$$
$$D_{hd} = 24h^2 + 2Nh$$
$$D_{dh} = D_g + D_{act} = 12h^2 + Nh$$

where $N$ denotes the total token number in the effective batch. $D_{hd}$, $D_{dh}$ stands for host-to-device and device-to-host data transfer, respectively. Meawhile for canonical approach, the model does not read the checkpoint activations during forward propagation. Hence, the host-to-device pattern can be listed as below:

$$D'_{hd} = \frac{N}{n}(24h^2 + nh)$$

Where $n$ is the maximal token number that accelerator can hold in one iteration. There we can have the elimination point of activation read overhead as mentioned in Sec. 2.2:

$$D_{hd} < D'_{hd}$$
$$N > \frac{24nh^2}{24h^2 - nh}$$

**Inference.** The case is slightly different in inference. There are severaly differences: **1.** Inference does not need to save activations for backward propagation, but **2.** with KV cache enabled, the K, V matrices of new token should be offloaded and reloaded each layer a sub batch is computed; 3. hidden states between layers is only 1-token long, therefore will not be offloaded. Thus, we have the I/O patterns listed below:

$$D_{hd} = D_p + D_{KV} = 12h^2 + 2Nhl_0$$
$$D_{dh} = D_{KV_i} = 2Nhl_1 \quad D'_{hd} = D_p = \frac{N}{n}12h^2$$

Where $D_{KV}$ refers to all past KV cache and $D_{KV_i}$ is the KV cache of the newly generated token. $l_0$ and $l_1$ are the lengths of prompt and newly generated tokens. Note here the $N$ and $n$ refers to effective batch size and maximum sub batch size. For vanilla approach, we do not offload and load anything other than model weights. Considering the fact that usually $D_{hd} >> Ddh$, we give a approximate estimation of overhead elimination by:

$$D_{hd} < D'_{hd}$$
$$N > \frac{12h^2}{12h^2 - 2nhl_0}$$

This result also suggests that enlarging effective batch size can hyperbolically hinder the overhead of fetching model weights in inference.

## A.2 NUMERICAL RESULTS OF SINGLE-DEVICE BENCHMARKING

| | **Ours** | **FlexGen** | **Vanilla** | | **Ours** | **Vanilla** | **Speed-Up** |
|---|---|---|---|---|---|---|---|
| **OPT-6.7B** | 112.52 | 74.26 | 38.39 | **OPT-6.7B** | 114.58 | 82.39 | 1.39 |
| **OPT-30B** | 47.50 | 22.73 | 5.51 | **OPT-30B** | 51.09 | 12.01 | 4.25 |
| **OPT-175B** | 3.01 | 1.28 | 0.13 | | | | |

Table 3: Single-device benchmarking results. **Left**: Inference (Token/s); **Right** Training (FLOPS)

