# OpenReview forum: "Balance Beam: adaptive computation for affordable training and inference with high-throughput offloading for LLMs"
_ICLR.cc/2024/Conference — ICLR 2024 Conference Withdrawn Submission_

### Official Review · Reviewer_ttTz · 2023-10-29

**Soundness:** 2 fair
**Presentation:** 1 poor
**Contribution:** 1 poor
**Rating:** 3
**Confidence:** 4

**Summary:**

The paper proposes a workflow to address the challenges of training and inference of LLMs. The workflow focuses on dynamic analysis of model-system compatibility and prioritizing computational intensity over data movement. The methods involve a hyperparameter selection strategy, and better offloading scheme.

**Strengths:**

1. The paper focuses on a critical and urgent issue, the training and inference efficiency of LLMs.
2. The paper provides an image on GCP to reproduce their partial results.

**Weaknesses:**

### Major issue 1
The paper "**theoretically**" analyzes the hyper-parameter selection process in Section 3.1 and provides experimental
validation in Section 5.1. However, Section 3.1 cannot demonstrate effective theoretical information.
1. The paper recommends a minimum batch size for achieving high throughput, and caching will not be employed if this threshold cannot be met. Is this threshold a hyper-parameter need to be configured? The threshold is definitely related to hardware specifications and model architectures. It is hard for other users to apply this technique. They have to tune this parameter empirically instead of configure it theoretically.
2. The method can select appropriate batch sizes by assessing the working memory requirements per token during benchmarking. It is unclear for readers the details of the selection.
3. The paper propose to to retain only one layer on the accelerator at any given time during inference, for checkpoint selection. Is this strategy guaranteed optimal theoretically? I think the setting of checkpoint depends on the hardware specifications and model architectures. I think the proposed is not optimal at all times.

Hyper-parameter balancing strategies in Section 3.1 is not a solid theoretical analysis. It seems a case study for a specific hardware/software settings and it is difficult to transfer to other settings. Users cannot follow their strategy to set hyper-parameters in an optimal way.

### Other major issues
1. The paper claims that Balance Beam is "an workflow to optimize the trade-off between latency and throughput performance of LLMs". However, there is no discussion on the latency in the proposed method and experiments. From my understanding, the column-wise traversal of Figure 1.a will increase the latency significantly compared to the row-wise traversal.
2. The evaluation results are based on the authors' implementations, for both baseline and the proposed method. However, the baseline may be much weaker than the current SOTA solution. Is it possible to import a public SOTA implementation and conduct comparisons based on that?
3. The paper use FLOPS to quantify the arithmetic intensity. FLOPS is a measure of computer performance, while arithmetic intensity is the ratio of total floating-point operations to total data movement. They do not match.

### Minor issues
1. It is not clear whether the proposed method can be applied to other hardware settings, such as other GPUs, TPUs and large scale.
2. the FlexGen proposed in (Sheng et al., 2023) have showed -> has shown
3. Table 2. Optimal -> Ours

**Questions:**

What are the limitations and potential negative impact of Balance Beam?

---

### Official Review · Reviewer_uQcc · 2023-11-01

**Soundness:** 3 good
**Presentation:** 1 poor
**Contribution:** 2 fair
**Rating:** 3
**Confidence:** 4

**Summary:**

The paper focuses on some hyperparameters ( KV cache, batch size, and #gradient checkpoints ) with a newly invented column-wise traversal approach. Via tuning those hyperparameters with some systematic optimization, this paper shows 2x speedups compared to baseline. This paper also proposes the approach can be applied to fine-training scenarios. Gets high acceleration on fine-tuning tasks.

**Strengths:**

The main strength of this paper is that the newly invented FlexGen approach can be applied to training scenarios, providing less data traffic. It has been impossible to scale batch size due to memory capacity in previous offloading training scenarios.

The paper introduces new hyperparameters into offloading, so tuning hyperparameters (KVcache, batch size, and gradient checkpoints) looks important for better utilization. For OPT-175B, turning off KVcache shows optimal performance, which is especially interesting.

 Providing real value for the experiments make it easy for comparing with different papers. In addition, if the codes are contributed to popular framework, it would be very helpful to apply common system optimization methods to derive higher throughput with newly invented methods.

**Weaknesses:**

I have concerns on aspects of novelty, practicality and evaluation.
### Novelty
- The main contribution of the paper is in providing a tunable hyperparameter space. However, the proposed hyperparameters are not newly discovered, but already existing knobs.
Personally, I find it interesting that the popularly used KV caching can be a new hyperparameter, and that putting it together with other parameters is a meaningful work.
However, I don't think the novelty is enough to be published in ICLR.

- The techniques provided from section 3.2 are mostly what's commonly used, especially in ZeRO-offload. Even though the paper does not strongly claim those as the contribution, it is not okay to introduce them in the methods section, without any citations.


### Practicality
- This paper suggests the batch size and sub batch size as the key hyperparameter.
However, unlike inference, changing the batch size has impact on the final accuracy. Because of this, it shouldn't be tuned just for the throughput.
In addition, I was a little disappointed that the paper just provides the results from various tuning points. At first, I was expecting a systematic/automatic tuning software or a policy, or at least a guideline on how to achieve the goal. Without them, the contribution is quite limited.

### Evaluation
- The authors seem to acknowledge that there is accuracy impact on using large batch sizes for training (from section 6.1), but there is no evaluation on the accuracy.

- Some experiments can be misleading. For training, ‘with optimal hyperparameter settings’ seems to use different effective batch sizes, and only FLOPS results are given. If effective batch size is different, the throughput difference would come from additional update steps, and this should be considered.


### Writing
- It's relatively minor compared to other issues, but the writing needs some improvements. For example, the paper does not clearly distinguish the training and inference. Even though they share a lot in common, some guide is needed for the readers. In addition, the description of generative inference with KV-cache is not enough to understand this work. Even though the KV cache is getting its popularity, this paper lacks enough information to understand the key aspects, such as why utilizing KV-cache is good for transformer inference.


There are some ambiguities and minor mistakes in this paper.
If Fig.1-(b) is placed next to Fig.3, it could be better to see with Sec.3.2.
In Sec.1, ‘5000’ and ‘s’ are placed in different lines, which makes it confusing.
In Sec.1, Insert ‘and’ between ‘batch size’ and ‘number of gradient checkpoints.’
I can’t find any references for gradient checkpoints and recompute.

**Questions:**

For evaluation, are training FLOPS results measured with different effective batch sizes?
For training, does this work utilize only host memory? (No storage for training? )

---

### Official Review · Reviewer_uduE · 2023-11-05

**Soundness:** 2 fair
**Presentation:** 2 fair
**Contribution:** 2 fair
**Rating:** 5
**Confidence:** 4

**Summary:**

The paper argues that the system hyperparameters are challenging to tune in a distributed setup and design and introduce a workflow to mitigate these challenges. The results show promising improvement over SOTA framework in inference as well as training.

**Strengths:**

$\mathtt{+}$ The results are promising and cover both inference and training of LLMs across a range of benchmarks.

$\mathtt{+}$ The framework seems to be straightforward to use.

**Weaknesses:**

$\mathtt{-}$ The contribution of the paper is limited and it is not clear whether the presented results are generalizable to other networks and setup.

$\mathtt{-}$ The majority of presented techniques are already explored and the paper mainly focuses on changing to system hyperparameters for a particular setup.

$\mathtt{-}$ The choice of the baseline is not well-supported and it is not clear whether the baseline implementation was also fully optimized for the target platform.

**Questions:**

(Q1) Did you also tune the system hyperparameters of the baseline?

(Q2) How does your approach scales to larger network (beyond 4 GPUs)?

(Q3) I am curious to see whether the proposed approach has any negative impact on accuracy/training convergence/etc.? Did you see any degradation?

(Q4) I commend the authors for devoting almost a page to future works. However, I think the suggested future work actually are really important to see the potential benefit of the work. In its current form, the paper seems to have a very narrow scope and no clarity on how to apply in different situations and scenarios.